

**The new Passive microwave Neural network Precipitation Retrieval (PNPR)**
**algorithm for the cross-track scanning ATMS radiometer: description and**
**verification study over Europe and Africa using GPM and TRMM spaceborne**
**radars**
Paolo Sanò[1], Giulia Panegrossi[1], Daniele Casella[1], Anna Cinzia Marra[1], Francesco Di Paola[2], Stefano
Dietrich[1]
[1] Institute of Atmospheric Sciences and Climate (ISAC), National Research Council of Italy (CNR), 00133 Rome,
Italy
[2] Institute of Methodologies for Environmental Analysis (IMAA), Italian National Research Council of Italy (CNR),
C.da S.Loja, Tito Scalo, 85050 Potenza, Italy
*Correspondence to:* Paolo Sanò (paolo.sano@artov.isac.cnr.it)



**Abstract.** The objective of this paper is to describe the development and evaluate the performance
of a totally new version of the Passive microwave Neural network Precipitation Retrieval (PNPR
v2), an algorithm based on a neural network approach, designed to retrieve the instantaneous
surface precipitation rate using the cross-track ATMS radiometer measurements. This algorithm,
developed within the EUMETSAT H-SAF program, represents an evolution of the previous version
(PNPR v1), developed for AMSU/MHS radiometers (and used and distributed operationally within
H-SAF), with improvements aimed at exploiting the new precipitation sensing capabilities of
ATMS with respect to AMSU/MHS. In the design of the neural network the new ATMS channels
compared to AMSU/MHS, and their combinations, including the brightness temperature differences
in the water vapor absorption band, around 183 GHz, are considered . The algorithm is based on a
single neural network, for all types of surface background, trained using a large database based on
94 cloud-resolving model simulations over the European and the African areas.
The performance of PNPR v2 has been evaluated through an intercomparison of the instantaneous
precipitation estimates with co-located estimates from the TRMM Precipitation Radar (TRMM-PR)
and from the GPM Core Observatory Ku-band Precipitation Radar (GPM-KuPR). In the
comparison with TRMM-PR, over the African area, the statistical analysis was carried out for a
two-year (2013-2014) dataset of coincident observations, over a regular grid at $0.5° \times 0.5°$
resolution. The results have shown a good agreement between PNPR v2 and TRMM-PR for the
different surface types. The correlation coefficient (CC) was equal to 0.69 over ocean and 0.71 over
vegetated land (lower values were obtained over arid land and coast), and the root mean squared
error (RMSE) was equal to 1.30 mm h$^{-1}$ over ocean and 1.11 mm h$^{-1}$ over vegetated land. The
results showed a slight tendency to underestimate moderate to high precipitation, mostly over land,
and overestimate moderate to light precipitation over ocean. Similar results were obtained for the
comparison with GPM-KuPR over the European area (15 months, from March 2014 to May 2015
of coincident overpasses) with slightly lower CC (0.59 over vegetated land and 0.57 over ocean)
and RMSE (0.82 mm h$^{-1}$ over vegetated land and 0.71 mm h$^{-1}$ over ocean), confirming a good
agreement also between PNPR v2 and GPM-KuPR. The performance of PNPR v2 over the African
area was also compared to that of PNPR v1. PNPR v2 has higher R over the different surfaces, with
general better estimate of low precipitation, mostly over ocean, thanks to improvements in the
design of the neural network and also to the improved capabilities of ATMS compared to
AMSU/MHS. Both versions of PNPR algorithm have shown a general consistency with the
TRMM-PR.





## 1. Introduction

The availability of data from the Advanced Technology Microwave Sounder (ATMS), a cross-track scanning radiometer currently onboard the Suomi National Polar-orbiting Partnership (Suomi NPP) satellite (and on the Joint Polar Satellite System (JPSS) series starting in 2017), represents an important step in short and long-term weather forecasting and environmental monitoring. Combining the capabilities of its predecessor sounders such as the Advanced Microwave Sounding Unit-A (AMSU-A) and the Microwave Humidity Sounder (MHS) aboard NOAA-18 and NOAA-19 and the ESA MetOp-A and MetOp-B satellites, ATMS provides sounding observations with improved resolution, sampling and coverage for retrieving atmospheric vertical temperature and humidity profiles. Moreover, this new-generation instrument provides more information about surface, vertical distribution of hydrometeors, precipitation, and other key environmental variables (Chen et al., 2007; Boukabara et al., 2013; Zou et al., 2013; Kongoli et al., 2015).

With regard to precipitation it should be mentioned that, although the reliable knowledge of its intensity and accumulation is essential for understanding the global hydrological and energy cycles, precipitation estimate (from satellite and from the surface) is complicated by several factors: the large variability of the precipitation in time and space, the conversion of satellite measurements into quantitative precipitation estimates, uncertainties associated to rain gauges (and to their spatial distribution), and radar measurements (i.e., attenuation, beam-blocking) and their unavailability in several regions in the world and over ocean (Mugnai et al., 1993; Iturbide-Sanchez et al., 2011; Bennartz and Petty, 2001; Tian et al., 2009; Kirstetter et al., 2012).

An important step forward towards the improvement of global precipitation monitoring is represented by the Global Precipitation Measurement (GPM) mission launched on 27 February 2014. GPM is expected to provide accurate precipitation estimates thanks to the availability of the NASA/JAXA GPM Core Observatory (GPM-CO) [equipped with the GPM Microwave Imager (GMI) and the Dual-frequency Precipitation Radar (DPR)], a common, global observatory of 3-D precipitation structure at 5 km resolution, and thanks to the exploitation of a constellation of international Low Earth Orbit (LEO) satellites equipped with microwave radiometers for precipitation observation, providing frequent measurements over most of the globe (3-hourly coverage between 65° S and 65° N) (Hou et al., 2014; Draper et al. 2014; Newell et al., 2014; Petkovic and Kummerow, 2015). A contribution of ATMS as part of the GPM constellation is foreseen in this direction, also in relation to the technological improvements over its predecessor sounders.

In Europe, the EUMETSAT "Satellite Application Facility on Support to Operational Hydrology and Water Management" (H-SAF, Mugnai et al., 2013a), has been called upon to participate in and to contribute towards the GPM by providing its own precipitation products, and being at the same time a user of GPM data and a direct collaborator of GPM on two main aspects: development and refinement of retrieval techniques through the exploitation of all available radiometers in the GPM constellation, and validation activity. In this context, operational passive microwave (PMW) precipitation products for the different radiometers are being released within H-SAF as new radiometers become available, and they are based on two approaches (Mugnai et al., 2013b): the physicaly-based Bayesian Could Dynamics and Radiation Databse (CDRD) algorithm (Casella et al., 2013, Sanò et al., 2013,) for conically scanning radiometers and the Passive microwave Neural network Precipitation Retrieval algorithm (PNPR) for cross-track scanning radiometers, originally developed for AMSU/MHS and fully described in Sanò et al. (2015) (PNPR-AMSU/MHS, hereafter PNPR v1).

The objective of this paper is to describe the development and evaluate the performance of a newly developed version of PNPR designed to retrieve the instantaneous surface precipitation using the ATMS radiometer data. This algorithm (PNPR-ATMS, hereafter PNPR v2) represents an evolution of PNPR v1 (used operationally within the EUMETSAT H-SAF) with improvements aimed at exploiting the new precipitation sensing capabilities of ATMS with respect to AMSU/MHS.



Neural Networks (NNs) represent a highly flexible tool alternative to regression and classification techniques, widely applied in an increasing fields of the meteorological research for their capability to approximate complex nonlinear and imperfectly known functions (e.g. Liou et al., 1999; Del Frate and Schiavon, 1999; Shi, 2001; Marzban, 2003; Blackwell and Chen, 2005; Chen et al., 2006; Krasnopolsky et al., 2008; Shank et al., 2008; Haupt et al., 2009; Aires et al., 2012).

NNs have been used in precipitation retrieval - precipitation being one of the most difficult of all atmospheric variables to retrieve - considering the opportunities offered by their ability to learn and generalize (Hsu et al., 1997; Hall et al., 1999; Staelin et al., 1999; Sorooshian et al., 2000; Chen and Staelin, 2003; Hong et al., 2004; Sussuravadee and Staelin, 2007, 2008a, 2008b, 2009, 2010; Bellerby, 2007; Krasnopolsky et al., 2008; Leslie et al., 2008; Mahesh et al., 2011). However, it should be mentioned that the use of NNs involves the training phase with a large representative database, often obtained from cloud-resolving model simulations. Consequently, the performance of the network is largely dependent on the completeness and the representativeness of the database and on its consistency with the observations.

Retrieval algorithms based on NNs, proposed for precipitation estimation from remotely sensed information, using MW or VIS/IR measurements, are different from each other in the different approaches used, in the design of the network architecture, in the selection of type and number of input variables, in the determination of the number of networks used, in the implementation of the training database  (e.g. the cloud-resolving model) and in the training process. With regard to the input variables, when MW radiometers are used, their choice is normally based on physical considerations on the radiometric signatures (or brightness temperatures, TBs) of different microwave channels and on the direct or indirect relationship of these signatures with environmental, meteorological and microphysical variables (e.g. atmospheric temperature and humidity, surface conditions, hydrometeor types, size and shapes) involved in the precipitation retrieval process. The TBs at the MW channels so identified are selected as part of the input variables. However, some techniques such as principal component analysis (PCA) are applied to the selected channels in order to reduce the number of inputs, and reduce the complexity of the NN, and also to reduce the noise (e.g., to filter out the signal due to the background surface), (Chen and Staelin, 2003; Surussavadee and Staelin, 2008a, 2010; Blackwell and Chen, 2005). Special functions of TBs, already proposed for rainfall retrieval (Kidd, 1998; Ferraro and Marks, 1995; Grody, 1991), as the Polarization Corrected Temperature ($PCT_{85}$) or the Scattering Index have also been considered as the NN inputs (Sarma et al., 2008; Mahesh et al., 2011). Some geographical and meteorological parameters (e.g. surface type, surface height, season, latitude) are often considered as auxiliary input data, in order to reduce the ambiguity intrinsic to the PMW precipitation retrievals based only on observed TBs (e.g., Panegrossi et al., 1998; Kummerow et al., 2011; You and Liou, 2012; You et al., 2015).

In the VIS/IR based NN algorithms the input selection is based on different considerations, due to the indirect relationship between cloud top radiances and surface rainfall and the lack of information on the precipitation structure within the cloud. Additional inputs are then considered in addition to TBs, that is, clouds texture information (TB mean and variance for 3x3 and 5x5 pixel rectangles around each measurement), the rate of change of cloud-top temperatures, the number of pixels with TB less than a given threshold (Hsu et al., 1997; Bellerby et al., 2000; Tapiador et al., 2004). This approach normally involves the use of more complex networks.

The number of NNs used in the precipitation retrieval algorithms, is defined so as to optimize the network performance under different operating conditions. In PMW precipitation retrieval separate NN algorithms are usually proposed depending on the type of surface (i.e., land or sea), to discriminate between the different precipitation emission signatures relative to background (e.g. Surussavadee and Staelin, 2008a). Separate NN algorithms are also proposed to deal separately with stratiform and convective precipitation (e.g Sarma et al., 2008).


In the design of PNPR v2 important aspects in relation to the topics mentioned above, concerning the choice of the inputs, the number of networks used by the algorithm, and the database used in the training phase, have been thoroughly analyzed and will be presented in this paper.

Another important issue to consider is that PNPR v2 has been designed in the perspective of the full exploitation of the MW radiometers in the GPM constellation of satellites, and of the achievement of consistency (besides accuracy) of the retrievals from the different sensors. These goals are considered priorities in the international GPM mission community because their achievement leads to a significant reduction of the errors, also associated with the inadequate sampling of precipitation, with positive impact on precipitation monitoring (see also Panegrossi et al., 2016), hydrological applications, and climate studies. This is also true when higher spatial/temporal resolution products based on MW/IR combined techniques are used, such as IMERG (GPM), TMPA (Tropical Rainfall Measuring Mission - TRMM), see Huffman et al. (2007, 2015), and also within the EUMETSAT H-SAF program, these aspects have become a priority. Therefore, PNPR v2 for ATMS, as well as PNPR v1 for AMSU/MHS, as well as all other H-SAF products for conically scanning radiometers, represent an important contribution towards the exploitation of the current and future constellation of PMW radiometers for global precipitation monitoring.

In this paper the PNPR v2 algorithm is described in detail, and the methodology and the results of an intercomparison of the PNPR v2 instantaneous precipitation estimates with co-located spaceborne radar estimates from the TRMM Precipitation Radar (TRMM-PR) and from the GPM-CO Ku-band Precipitation Radar (GPM-KuPR) are presented.

Section 2 presents a brief description of the characteristics of ATMS. In Section 3 a description of the PNPR v2 algorithm is presented, with reference to the design of the neural network, the main characteristics of the algorithm, and the relevant features of the ATMS training database. The verification study is presented in Section 4, which includes a brief description of the characteristics of PR and DPR, of the methodology used to create the co-located observation dataset used in the study, the analysis of the performance of PNPR v2 compared to TRMM-PR and to GPM-KuPR, and a comparison with PNPR v1 using TRMM-PR rainfall estimates as reference. Section 5 contains the conclusive remarks about the performance of PNPR v2 and future perspectives.

## 2. The ATMS radiometer

ATMS is a total power cross-track scanning microwave radiometer on board the Suomi National Polar-orbiting Partnership (NPP) satellite (and JPSS satellites scheduled for early 2017), with a swath of 2600 km, angular span of ±52.77° relative to nadir (Boukabara et al., 2011, 2013; Weng et al., 2012; Goldberg et al., 2013; Zou et al. 2013). During each scan the Earth is viewed at 96 different angles, with a spatial sampling of 1.11°. ATMS has 22 channels, ranging from 23 to 183 GHz, providing both temperature soundings from the surface to the upper stratosphere (about 1 hPa, ~45 km), and humidity soundings from the surface to upper troposphere (about 200 hPa, ~15 km). Particularly, ATMS channels 1–16 provide measurements at microwave frequencies below 60 GHz and in an oxygen absorption band, and channel 17–22 are located at higher microwave frequencies above 89 GHz and in a water vapor absorption band. The beamwidth changes with frequency and is 5.2° for channels 1-2 (23.8-31.4 GHz), 2.2° for channels 3-16 (50.3 - 57.29 GHz and 88.2 GHz), and 1.1° for channels 17-22 (165.5-183.3 GHz). The corresponding nadir resolutions are 74.78 km, 31.64 km and 15.82 km respectively. The outmost FOV sizes are 323.1 km x 141.8 km (cross-track x along-track), 136.7 km x 60.0 km, 68.4 km x 30.0 km, respectively.

Compared with its predecessors AMSU and MHS, ATMS has improved resolution (31.6 km at nadir in the 54 GHz band, vs 48.6 km for AMSU) and spatial sampling (1.11° in the 54 GHz band, vs 3.33° for AMSU) and has the great advantages of a wider swath that practically eliminates the orbital gaps. There are slight differences in the frequencies of ATMS channels 88.2 GHz, 165.5 GHz and 183.31 ± 7.0 GHz with respect to the corresponding MHS channels (89.0 GHz, 157.0 GHz and 190.31 GHz). Three new channels are added compared to AMSU/MHS: channel 4 (51.76 GHz) for lower tropospheric temperature sounding, and the two channels 19 and 21 (183.31±4.5 GHz and





183.31±1.8 GHz) to enhance the moisture profiling performance, improving the vertical resolution, and potentially very useful also for precipitation (Surussavadee et al., 2012; Weng et al., 2012; Zou et al., 2013).

## 3. The new PNPR algorithm

### 3.1 Algorithm description

PNPR v2 represents an evolution for ATMS applications, of the previous PNPR v1 algorithm based on a NN approach, developed at ISAC-CNR for precipitation rate estimation using AMSU/MHS observations. The full description of PNPR v1 is provided in Sanò et al. (2015), while some important aspects are reviewed in this paper for completeness.

Both versions of PNPR are designed to work over the full Meteosat Second Generation (MSG) disk area (60°S-75°N, 60°W-60°E). In PNPR v1 the training of the NN was carried out using two distinct NNs, one for the European/Mediterranean area (Sanò et al., 2015) and one for the African area (Panegrossi et al., 2014). Each network was designed to work with all types of surface backgrounds (i.e., land, sea, coast) in order to reduce the discontinuity of precipitation estimates often found in correspondence with transitions between surfaces with different radiometric properties. In PNPR v2 one unique NN has been designed, capable of operating on the whole MSG disk area, regardless of the type of surface and of the geographical area.

Another significant aspect in the design of PNPR v1 was the use of the TB differences in the water vapor absorption band channels at 183 GHz as input to the neural network. Opaque channels around 183 GHz were originally designed to retrieve water vapor profiles due to their different sensitivity to specific layers of the atmosphere (Wang et al., 1997; Staelin and Chen, 2000; Blackwell and Chen, 2005). However these channels have shown great potentials for precipitating cloud characterization and for precipitation retrieval. The different penetration ability of these channels in the atmosphere can be exploited to analyze the vertical distribution of hydrometeors (Wang et al., 1989, 1997; Burns et al., 1997; Staelin and Chen, 2000; Ferraro et al., 2005; Hong et al., 2005, 2008; Funatsu et al., 2007, 2009; Laviola and Levizzani, 2011), and to obtain some criteria for the characterization of precipitation as weak, moderate, strong convective or stratiform, using the TB differences $\Delta_{17}$, $\Delta_{13}$, and $\Delta_{37}$ (corresponding respectively to the differences between the 183.31±1 GHz and 183.31±7 GHz, 183.31±1 GHz and 183.31±3 GHz, 183.31±3 GHz and 183.31±7 GHz channels) (e.g., Ferraro, 2004; Qiu et al., 2005). In the design of PNPR v2 we have focused on the exploitation of the improved technical characteristics of ATMS with respect to AMSU/MHS, with the analysis of the information carried by two new channels in the 183 GHz water vapor absorption band (at 183.31±4.5 GHz and 183.31±1.8 GHz) (see Section 3.3 dedicated to the input selection). As in PNPR v1, a Canonical Correlation Analysis has been carried out to find the linear combination of TBs of selected channels best correlated with surface precipitation rate.

The flow diagram of the PNPR v2 algorithm is substantially the same as that of PNPR v1, described in detail in Sanò et al. (2015), except for the use of one unique network trained on a database representative of MSG full disk area (see Section 3.2), and changes in the input selection in the design of the network (described in Section 3.3). Furthermore, in the preprocessing of the brightness temperatures, in addition to the decoding of the file format and the quality control of the input data, the removal of the three outmost pixels along the scan is carried out. Other processing steps of the algorithm, such as the screening procedure of no-rain pixels, the quality index map providing indications on areas or conditions where the retrieval is more or less reliable, are unchanged with respect to those used for the algorithm PNPR v1 (Sanò et al., 2015). In a similar way, also the new algorithm provides in output, in addition to the precipitation rate (mm/h), the phase of the precipitation (solid, liquid, mixed or unknown), and the quality index. The PNPR v2 output is provided on a grid corresponding to the ATMS nominal resolution varying from 15.82 km x 15.82 km / circular at nadir to 68.4 km x 30.0 km / elliptical at scan edge.





## 3.2 The training database

The training of PNPR v2 was performed using a large cloud-radiation database representative of the MSG full disk area, built from 94 cloud-resolving model (CRM) simulations of different precipitation events including 60 simulations over the European/Mediterranean area (Casella et al., 2013), and 34 simulations over Africa and Southern Atlantic (Panegrossi et al, 2014). The simulations were carried out using the University of Wisconsin Nonhydrostatic Modeling System (UW-NMS) (Tripoli, 1992; Tripoli and Smith, 2014a, b)) coupled to a Radiative Transfer Model (RTM) relating CRM environments to expected top-of-atmosphere PMW TBs of the ATMS radiometer (see Smith et al., 2013, and Casella et al., 2013 for the details about the cloud model configuration setup, and Sanò et al., 2015 for AMSU/MHS RTM simulations). Figure 1 shows the geographical location of the inner domain of the 94 simulations. Simulated events were selected in order to cover the different seasons and different meteorological situations and precipitation regimes. Simulations over African and South Atlantic area were chosen also on the basis of the TRMM-PR observations (in particular the Rain Type flag and the Freezing level height) and on the basis of different climatic regions in order to cover as much as possible the climatic variability in the area of interest with a limited number of simulations.

The simulated TBs were calculated considering the different ATMS viewing angles and channel frequencies using the same approach used for AMSU/MHS and described in Sanò et al., 2015. The database contains more than two million entries for the European/African regions and has 45 views for each entry (the three outmost pixels were discarded due to the low resolution).

## 3.3 Input selection

The first objective in the new NN design was the selection of the inputs based on the evaluation of their impact on the performance of the NN or on their sensitivity to precipitation. Consistently with PNPR v1 and on the basis of the results obtained for AMSU/MHS (Sanò et al., 2015), for the new NN we have initially imposed the use of the three inputs $\Delta_{17}$, $\Delta_{13}$, and $\Delta_{37}$ (Hong et al., 2005; Funatsu et al., 2007, 2009). These TB differences have been proven to be very effective in detecting precipitation, differentiating between different precipitation structures and in the retrieval of rainfall rate. For PNPR v2, a detailed analysis to evaluate the effect of additional inputs on the performance of the NN has been carried out. All possible TB differences with the two new ATMS 183 GHz (183.31±1.8 GHz and 183.31±4.5 GHz) channels were considered, and the analysis was based on a cross-validation method (Anders and Korn, 1999; Marzban, 2009), already used for PNPR v1 (Sanò et al., 2015). This method consists, essentially, in comparing the quality of two NNs by evaluating their mean squared prediction error (MSPE) when they are applied to an equal number (M) of validation data sets. Therefore, the cross-validation index (CV) is defined as:

$$CV = \frac{1}{M} \sum\nolimits_{m=1}^{M} MSPE_m$$

In a first test, only the three differences $\Delta_{14}$, $\Delta_{24}$, and $\Delta_{27}$ (corresponding respectively to the differences between the 183.31±1 GHz and 183.31±4.5 GHz, 183.31±1.8 GHz and 183.31±4.5 GHz, and 183.31±1.8 GHz and 183.31±7 GHz channels) showed a real improvement in the NN performance. The use of the differences between contiguous channels resulted in fact irrelevant. The subsequent tests with these three new inputs proved that $\Delta_{24}$, added to $\Delta_{17}$, $\Delta_{13}$, and $\Delta_{37}$ already selected in PNPR v1, was the input with most significant impact on the NN performance. Table 1 shows some results obtained during the test.

In the table the various possible differences considered as input to the NN in this analysis are shown in the first column; $\Delta_F = \Delta_{13}, \Delta_{37}, \Delta_{17}$ denotes the three difference combinations used in the PNPR v1 algorithm. In the second and fourth columns the values of the correlation coefficients between output and target during the learning phase ($R_L$), and the mean values during the cross-validation





phase ($R_{CV}$) are shown. In the third and the fifth columns the values of the mean squared error during the learning phase ($MSE_L$), and the cross-validation index CV (the mean MSPE values during the cross-validation phase) are provided. From the results shown in the table it is evident that the NN performance improves when the input $\Delta_{24}$ is added. It is worth noting that to achieve the results shown in Table 1 the training protocol described in Sanò et al. (2015) has been applied, and that for each input configuration (each row in the table) more than one hundred NNs (with different levels of perceptrons) were compared to select the optimal network configuration.

The contribution of $\Delta_{24}$ as new input can be seen as a compensation of $\Delta_{17}$ when, under certain conditions, this is affected by the "noise" of the background surface. In fact, the 183±7 GHz channel, the most penetrating among the 183 GHz channels, has a weighting function peaking at the lowest levels (Bennartz and Bauer, 2003), and the TB can be significantly affected by the signal from the underlying surface (for example in cold and dry conditions). On the other hand, in the same conditions, the 183±4.5 GHz channel has a weighting function peaking at higher levels. Some tests have been carried out on different cloud model profiles extracted from the CRM simulations and based on RTM computations to analyze the behavior of the $\Delta_{17}$ and $\Delta_{24}$, and they have confirmed these effects (not shown). We have also verified that, by replacing $\Delta_{17}$ with $\Delta_{24}$, a lower performance of the network is achieved, whereas the combined use of the two differences guarantees the optimal performance. The use of $\Delta_{24}$ as added input to $\Delta_{17}$, $\Delta_{13}$, and $\Delta_{37}$ in PNPR v2 represents the best compromise between the achievement of a good performance and the minimization of the number of inputs of the NN in order to reduce its complexity, key aspects in any NN design.

Another difference between PNPR v2 and PNPR v1 algorithms is the result of the canonical correlation analysis (CCA) applied to the training database to find the linear combination of TBs (LCT) of selected channels best correlated with surface precipitation rate, to be used as additional input to the network (see Sanò et al., 2015). The resulting linear combination for ATMS is composed of the window channels 31.4 GHz, 88.2 GHz, and 165.5 GHz, showing the highest correlation coefficients in the CCA analysis (with respect to the surface rain rate) for all types of background surfaces (in PNPR v1 for AMSU/MHS the 50.3 GHz, 89 GHz, and 150 GHz were selected for LCT).

With regard to other inputs to the network, in PNPR v2 the same ancillary data used in PNPR v1 were maintained (surface height, background surface type, month, and secant of the zenith angle along the ATMS cross-track scan). An additional auxiliary input was added to drive NN in the transition between the European and African area, i.e., the monthly mean total precipitable water (TPW) obtained from ECMWF Era Interim reanalysis in the 2011-2014 period. It should be mentioned that the use of geographical and environmental/meteorological parameters (including TPW) in PMW precipitation retrieval is utilized to reduce the ambiguity intrinsic to the PMW precipitation retrieval process (for example in the NASA GPM Bayesian algorithms - see Kummerow et al., 2011, 2015; Kidd et al., 2016).

During the phase of network design and the training process, more than 400 architectures have been tested and an optimal NN has been obtained, where "optimal" refers to the one with best performance, i.e., minimum CV over the full dynamic range of the inputs, absence of overfitting, and absence of anomalous inhomogeneities in the retrievals (Sanò et al., 2015; Staelin and Surussavadee, 2007).

In summary, ten input variables (five TBs derived and five ancillary inputs) are used in the NN for ATMS:

1. a linear combination of TBs (LCT) at 31.4, 88.2 and 165.5 GHz;
2. $\Delta_{17}$ difference between the TBs of channels 183.31±1 and 183.31±7 GHz;
3. $\Delta_{37}$ difference between the TBs of channels 183.31±3 and 183.31±7 GHz;
4. $\Delta_{13}$ difference between the TBs of channels 183.31±1 and 183.31±3 GHz;
5. $\Delta_{24}$ difference between the TBs of channels 183.31±1.8 and 183.31±4.5 GHz;
6. surface type (land, sea, coast);





7. monthly mean TPW;
8. month;
9. surface height (altitude);
10. secant of the zenith angle.
The network architecture is similar to that of PNPR v1, with one input layer (with number of nodes
equal to the number of inputs) and two hidden layers with 23 and 10 nodes in the first and in the
second layer respectively (the number of nodes differs from PNPR v1). The tan-sigmoid transfer
function is used for the input and the hidden layers, while a linear transfer function is used for the
output node.
**3.4 Sensitivity analysis**
During the training procedure, an assessment of the sensitivity of the NN output to variations of the
inputs was carried out. Sensitivity analysis provides an estimation of the relative importance of the
inputs (Coulibaly et al., 2005). The knowledge of the NN behavior, in relation to input perturbation,
helps to assess the relevance of the individual contributions to the output, and to verify the correct
training of the NN (i.e., the weights remain stable) that is achieved when there is no significant
changes of the sensitivity during the last training iterations (epochs).
The sensitivity analysis, limited to the TBs derived variables that are more related to the rain rate
estimate, and not to the ancillary variables, was applied to the "optimal NN" (i.e., defined by the
listed inputs and the architecture described in the previous section), and was carried out during the
final phase of the training (see Sanò et al., 2015). The final phase was reached when the two
parameters indicating the quality of the learning process, i.e. the correlation coefficient (R) and the
gradient of performance (mean squared error), were respectively larger than 0.89 and less than 0.05,
with the number of epochs in the 700-900 range (see Sanò et al., 2015 for more details on this
procedure). The assessment of the sensitivity was carried out several times, in correspondence to
successive epochs (to ensure the representativeness of the data used for the analysis), and for three
different surface types (land, coast and ocean), using NN input data randomly extracted from the
training and test databases. Five inputs ($\Delta_{13}$, $\Delta_{37}$, $\Delta_{17}$, $\Delta_{24}$, LCT) were slightly perturbed by
percentages of their value within three times their standard deviation (calculated in the database).
The relative sensitivity (S) of the NN to each input (i.e., for a number of input perturbations) is
calculated as the ratio between the mean standard deviation of the output (i.e., the surface rainfall
rate), and the mean standard deviation of the input.

$$S_i = \frac{\overline{\sigma(RR_i)}}{\overline{\sigma(V_i)}}$$

where $S_i$ is the relative sensitivity corresponding to the input $V_i$ and $\sigma(RR_i)$ and $\sigma(V_i)$ are the
standard deviations of the rainfall rate and the input variable. Figure 2 shows the results obtained
for the three different background surface types considered.
The results show a similar behavior of the sensitivity for the three different surface backgrounds
considered. It is evident the higher sensitivity of NN with respect to the LCT in comparison with the
other inputs; this is due to the contribution of window channels used in LCT, selected by
maximizing the correlation with the surface precipitation rate. Another important aspect is the
relative contribution of the other inputs (TBs difference in the 183 GHz band channels) quite
similar among the three types of surface, with a slightly higher contribution of the input $\Delta_{17}$ for land
and coast and a good contribution of the new ATMS input $\Delta_{24}$ for all surface types.
**4.  Verification study**
**4.1 Dataset description**



This section presents the verification study carried out for the PNPR v2 algorithm, using as
reference the data provided by the TRMM and GPM spaceborne radars. The TRMM-PR is a 13.8
GHz radar with a swath width of 247 km (after the satellite was boosted to higher orbit in 2001). Its
coverage allows regional intercomparison of convective–stratiform contributions to precipitation
across the Tropics, with data available since the launch of the satellite in November 1997 until
October 2014. It is considered the precursor to GPM DPR, and has represented, during this time
interval, the best available remote-sensing instrument for precipitation (Schumacher and Houze,
2003). The TRMM PR2A25 product (Iguchi et al. 2000) provides rainfall rates based on the
reflectivity-rainfall rate relationships, along with a raindrop size distribution (DSD) model,
attenuation correction, and a non-uniform beam filling correction. Even though issues have been
raised about the accuracy of PR2A25, related to surface properties, variations of the DSD, or impact
of incidence angles (i.e., Iguchi et al., 2009; Hirose et al., 2012; Kirstetter et al., 2013), during its
operational period this radar has provided accurate estimates of instantaneous rain rate, as well as
calibration for other precipitation-relevant sensors in sun-synchronous orbits (Bellerby et al., 2000;
Heymsfield et al. 2000; Liao et al. 2001; Schumacher and Houze, 2003; Lin and Hou, 2008). The
GPM DPR (on board the GPM-CO) is composed of two precipitation radars, the GPM-KuPR  at
13.6 GHz (an updated version of the TRMM-PR), and the Ka-band Precipitation radar (GPM-
KaPR) at 35.5 GHz. The simultaneous use of the two radars was designed to obtain a greater
dynamic range in the measurements, more detailed information on the microphysical rain structure
(such as raindrop size distribution), and a consequent better accuracy in the rainfall retrieval (Le and
Chandrasekar, 2103a, 2013b; Hou et al., 2014; Chandrasekar et al., 2014). KuPR and KaPR have
the same space resolution at nadir, equal to 5.2 km, the same beamwidth, equal to 0.71°, and cross
track swath widths of 245 km and 120 km, respectively.  In this study we have considered only the
GPM-KuPR products because of the similarity with the TRMM-PR and because its larger swath
size offers better chances to find coincident observations with ATMS.  It is worth considering also
that in spite of the similarity between the two radars, the GPM-KuPR has higher sensitivity (with
minimum detectable reflectivity between 12 dBZ and 14 dBZ, outperforming the original
instrumental design of 18 dBZ) (Toyoshima et al., 2015; Hamada and Takayabu, 2016) than the
TRMM-PR radar (18 dBZ minimum detectable reflectivity).
Two datasets have been created, one composed by two years (2013-2014) of coincident Suomi-NPP
ATMS and TRMM-PR overpasses over the African area (36° S - 36° N and 60° E - 30° W), and
one made of 15 months (1 March 2014- 31 May 2015) of ATMS and GPM-KuPR coincident
overpasses over the European and African areas (36° S - 65° N and 60° E - 30° W). In the study the
comparison is carried out between the PNPR v2 precipitation rate and the NASA/JAXA
precipitation products from the two spaceborne radars, in particular the TRMM-PR standard
product 2A25 (V7), and GPM 2ADPR Ku normal scan (Ku-NS) (V03).  Coincident observations in
the area of interest within a 15 minute time window have been considered between ATMS and
TRMM-PR (hereafter ATMS-PR) and between ATMS and GPM-Ku-NS (hereafter ATMS-DPR-
Ku).
It should be pointed out that the results obtained from the ATMS-DPR-Ku coincidence dataset are
not as robust as the results obtained from the ATMS-PR dataset because of the limited size of the
dataset, and because of some uncertainties in the less consolidated day-1 V03 DPR products, linked
to factors such as the DSD parameterization (Liao et al., 2014), the evaluation of the path-integrated
attenuation (PIA), the surface reference technique (SRT), and the non-uniform beam filling effect
(NUBF) (Shimozuma and Seto, 2015).
Figure 3 (left panel) shows the geographical distribution (on the ATMS grid) of about 1.8 milions
coincident pixels ATMS-PR found over the African area in the two-year time frame 2013-2014.
The figure shows a rather good coverage of the entire area, with a number of coincident pixels
between 30 and 150 on Central Africa, increasing moving to the North and to the South.
In the right panel of the figure, the distribution of the coincident pixels ATMS-DPR-Ku over the
European and African areas, between March 2014 and May 2015, is shown. In contrast to the left





panel, the coverage is not as good with a lower number of coincident pixels, and with some uncovered areas. The number on coincident pixels increases over northern Europe at the high latitudes reaching a maximum value around 200. In the southern part of Europe and Africa, the number of coincidences is significantly reduced (maximum values around 50).

To obtain co-located vectors of rainfall estimates of ATMS and TRMM-PR, and of ATMS and GPM-KuPR, the radar precipitation rate at the surface was downscaled to the PNPR v2 product nominal resolution (variable along the scan line, see Section 3.1), by averaging the rainfall rate of all radar pixels falling within each PNPR v2 pixel. In order to reduce the geolocation and synchronization errors, due to the different viewing geometry of ATMS and the spaceborne radar, and to the time lag between the observations, statistical analysis was carried out over a regular grid at $0.5° \times 0.5°$ resolution. For some of the analysis the coincidence datasets were categorized on the basis of the background surface – vegetated land, arid land (for Africa only), ocean, and coast – using a digital land/sea map at 2s of arc resolution (see Casella et al., 2015).

## 4.2 Comparison with TRMM-PR

Figure 4 shows the geographical distribution of the values of three statistical indexes (hit bias, correlation coefficient (CC), and Root Mean Squared Error (RMSE), (see Tian et al. (2016) for the definition of these scores), obtained for the ATMS-PR dataset. The scores are computed considering all coincident ATMS-PR pixels within each $0.5° \times 0.5°$ grid box (regardless of the time of the overpasses) with precipitation rate greater than 0 mm h$^{-1}$ both from the radiometer and the radar  (hits only).

The top panel shows a rather uniform distribution of low bias (between -0.2 and 0.1 mm h$^{-1}$, negative in most regions), with areas with larger positive bias (0.8 mm h$^{-1}$) over the Equatorial region, mostly over the Atlantic and Indian Ocean, and a few scattered areas of larger negative bias (-0.8 mm h$^{-1}$). Moreover, the algorithm shows an overall good correlation (middle panel) (CC > 0.8 in most areas), and a RMSE (bottom panel) with a pattern quite similar to the hit bias, with most values between 0.2 and 0.5 mm h$^{-1}$, and a limited number of grid points with values around 1.3 mm h$^{-1}$. Overall, the panels point out a good agreement between PNPR v2 and TRMM-PR, evidenced by the widespread low values of bias and RMSE and the high values of CC.

In Fig. 5 the density scatterplots for all $0.5° \times 0.5°$ resolution grid-boxes of the ATMS-PR dataset are shown for different surface types. In the scatterplot, the coordinates are the values (in logarithmic scale) of the mean precipitation rate from ATMS and for TRMM-PR in each grid-box, while the color represents the number of points in the dataset for each pair of precipitation rate values. The correlation is quite good for all background surfaces. A significant number of coincident observations below the diagonal is found over ocean, mostly for precipitation rates less than 1 mm h$^{-1}$, and over vegetated land (for all precipitation rates) This confirms the overall slight underestimation (negative hit bias) over land of Fig. 4 (top panel). The values of the statistical indexes (hit bias, CC and RMSE) calculated over the entire dataset are also provided, and they confirm the good agreement between PNPR v2 and TRMM-PR for the different surface types (results for coast and arid land are affected by the low number of coincident pixels found for these areas). The (small) bias is negative for vegetated land (-0.08 mm h$^{-1}$) and arid land (-0.05 mm h$^{-1}$), and positive for ocean and coast (0.05 mm h$^{-1}$). Low RMSE is also found for all surface types, higher for ocean (1.30 mm h$^{-1}$), than for vegetated land (1.11 mm h$^{-1}$), and equal to 0.80 and 1.37 mm h$^{-1}$ for arid surfaces and coastal area respectively.  CC is higher for vegetated land (0.71), compared to ocean (0.69), coast (0.65) and arid land (0.64).

Table 2 presents the contingency table for the ATMS-PR dataset, based on the mean rainfall rate from ATMS and TRMM-PR within each $0.5° \times 0.5°$ grid-box.  The percentages shown in a given column, provided for the four surface backgrounds, represent how the PNPR v2 product classifies the precipitation assigned to each TRMM-PR class. Four rainfall rate intervals were selected in this comparison, 0.01 - 0.25 mm h$^{-1}$, 0.25 - 1 mm h$^{-1}$, 1 - 5 mm h$^{-1}$ and 5 - 15 mm h$^{-1}$. There is an appreciable general consistency between PNPR v2 and TRMM-PR estimates, as shown by the





largest percentages found on the main diagonal for each type of surface background. The
percentages exceed 70% for low precipitation rates ($\leq 0.25$ mm h$^{-1}$) and 50% for higher
precipitation rates. Looking at the distribution of the percentages for each radar range (in each
column), it is noticeable the underestimation of PNPR v2 compared to TRMM-PR (higher
percentages in the cells above the diagonal), which confirms what shown in Fig. 5, for vegetated
land and ocean.
Table 3 shows the Performance Index calculated for the different background surfaces, defined as:

$$Perf.Index = 100 * \frac{\sum_{i=1}^{4} \frac{n_{ij(i=j)}}{\Delta_i}}{\sum_{i=1}^{4} \left( \frac{\sum_{j=1}^{4} n_{ij}}{\Delta_i} \right)}$$

where $n_{ij}$ is the number of occurrences in cell ij, i is the column index and j is the row index, and $\Delta_i$
is the width of the i-th rain rate class (mm h$^{-1}$). For each surface type, the index consists of the
weighted sum of the number of occurrences in the main diagonal of the cells, divided by the
weighted sum of the total number of occurrences, where the weight is the rain rate range for each
class. The values shown confirm the good ability of the PNPR v2 to provide precipitation rates
consistently with TRMM-PR, mostly over ocean.

**4.3 Comparison with GPM-KuPR**
As mentioned previously, a verification of PNPR v2 algorithm has been made also using
precipitation rate estimates from the GPM-KuPR, available at mid-high latitudes. This was initially
intended for the European area only, where a larger number of coincident overpasses are available
during the time frame considered (March 2014 - May 2015) (see Fig. 3). However, results are
shown also for the African area, despite the lower number of coincidences available, in order to
assess the degree of consistency of the results obtained over the same area with the two Ku-band
spaceborne radars.
As for the comparison with the TRMM-PR, all co-located ATMS and GPM-KuPR retrievals were
regridded at a 0.5° × 0.5° resolution, and only grid-boxes with precipitation rates greater than 0 mm
h$^{-1}$ (hits) are considered. In Fig. 6 the density scatterplots over the African area and the European
area are shown, for vegetated land and ocean (the number of coincidences for the other surface
types are too low in the ATMS-DPR-Ku dataset). The corresponding values of bias, CC and RMSE
computed over the whole dataset are also provided in each panel. Over Africa, both panels show
patterns quite similar to those found in the comparison with TRMM-PR (Fig. 5). Also in this case
there is a slight underestimation for low precipitation (< 1 mm h$^{-1}$) more evident over the ocean.
Table 4 shows the comparison between statistical indexes obtained for the ATMS-DPR-Ku dataset
and those obtained for the ATMS-PR dataset, shown in Fig. 5.
The Table shows a good agreement between the scores obtained with two datasets, with very low
bias (slightly positive/negative over land/ocean for the ATMS-DPR-Ku dataset, while the reverse is
valid for the ATMS-PR dataset), low RMSE (lower for the ATMS-DPR-Ku dataset), and good
correlation.
The right panels of Fig. 6 show the scatterplots obtained in the comparison of PNPR v2 and GPM-
KuPR over the European area. Pixels with likely presence of ice or snow on the ground have been
eliminated from the dataset, in order to exclude from the verification study cases of snowfall (or
precipitation over frozen background), whose precipitation rate estimate is affected by larger
uncertainty (both in the GPM-DPR-Ku V03 product and in PNPR). For the identification of these
pixels the "Snow Depth" and "Sea Ice Cover" products from the ECMWF Era Interim re-analysis
(at 0.5° × 0.5° resolution) available every 6 hours have been used. A dataset corresponding to the
ATMS-DPR-Ku coincidence rainfall pixels has been created, considering the ECMWF re-analysis
closest in time to each overpass, and using the nearest-neighbor approach to match the ATMS-
DPR-Ku pixels with the ECMWF grid.
The scatterplots in Fig. 6 show a similar behavior for vegetated land for the two areas, while over
ocean in the European area there is a general tendency of PNPR v2 to overestimate the precipitation
with respect to the GPM-KuPR. The total bias has very low values, negative for vegetated land (-
0.12 mm h$^{-1}$) and positive for ocean (0.12 mm h$^{-1}$). The CC show lower values than for the African
region for the two background surfaces (0.59 for vegetated land, 0.57 for ocean), while the RMSE is
lower than over the African region, equal to 0.82 mm h$^{-1}$ for vegetated land, equal to 0.71 mm h$^{-1}$
for ocean.
In order to better interpret the results in Fig. 6, the geographical distribution over the European area
of bias, CC and RMSE is shown in Fig. 7 (similarly to Fig. 4). In these maps the statistical indexes
are evaluated including pixels with snow or ice on the ground. There is a prevalence of a positive
bias (although mostly below 0.3 mm h$^{-1}$, with some peaks above 0.5 mm h$^{-1}$) over the ocean (in the
Northern Atlantic Ocean, top-left panel) and in the few areas available in the coincidence dataset
over the Mediterranean Sea. In the remaining areas and over land there is a rather uniform
distribution of lower bias (between -0.2 and 0.2 mm h$^{-1}$). The correlation CC has quite high values
(prevalently between 0.80 and 1) throughout the European area, except for some regions in the
Northern Atlantic Ocean, where the values are around 0.6. The RMSE presents quite similar
patterns as the hit bias, with higher values (around 0.7 mm h$^{-1}$ and 1-1.5 mm h$^{-1}$ for few pixels)
where the bias is high, and lower values (less than 0.5 mm h$^{-1}$) where the bias is low.
**4.4 Comparison with PNPR v1**
In the second part of the verification study we have compared the performances over the African
area of the PNPR v2 with the PNPR v1,, to evaluate whether the use of the new ATMS channels
and the newly designed NN have led to improvements in the retrievals. The performance of the
PNPR v1 algorithm has been tested on the same two years period (2013-2014) used for PNPR v2,
considering coincident observations of AMSU/MHS radiometers, on board the NOAA-18, NOAA-
19, Metop-A and Metop-B satellites, with TRMM-PR. The PNPR v1 and TRMM-PR coincidence
dataset is made of about three million pixels.  The procedure used to evaluate the PNPR v1
performance is the same as that adopted for the PNPR v2 algorithm, described in section 4.1.
Table 5 presents the values of the statistical indexes hit bias, CC, and RMSE obtained in the
comparison of PNPR v1 and PNPR v2 with TRMM-PR precipitation retrievals, over a 0.5° × 0.5°
regular grid, for different background surfaces. These results indicate a good agreement of both the
algorithm retrievals with the TRMM-PR (NASA/JAXA product 2A25), and a better performance of
PNPR v2 especially over vegetated land, in terms of CC (0.71 for PNPR v2 vs. 0.68 for PNPR v1)
and RMSE (1.11 mm h$^{-1}$ vs. 1.65 mm h$^{-1}$), and over ocean in terms of all scores (CC = 0.69 vs. 0.61,
RMSE = 1.30 mm h$^{-1}$ vs. 2.32 mm h$^{-1}$ and hit Bias = 0.05 mm h$^{-1}$ vs. 0.59 mm h$^{-1}$). Over arid land
and coast PNPR v2 results might be affected by the limited size of the ATMS-PR coincidence
dataset (ATMS is on board one satellite only, while AMSU/MHS is on board four different
satellites). Improvements compared to PNPR v1 are evident in terms of hit bias (-0.05 mm h$^{-1}$ vs.
0.30 mm h$^{-1}$ for arid land, and 0.05 mm h$^{-1}$ vs. 0.20 mm h$^{-1}$ for coast), and in terms of RMSE (0.80
mm h$^{-1}$ vs 1.84 mm h$^{-1}$ for arid land, and 1.37 mm h$^{-1}$ vs 1.90 mm h$^{-1}$ for coast).
A further analysis of the performance of the two algorithms has been performed through the study
of the Relative Bias percentage (RB$_\%$) and the Adjusted Fractional Standard Error Percentage
(AFSE$_\%$) used to remove systematic errors (Tang et al., 2014), as a function of the mean TRMM-PR
rainfall rate value computed for different rainfall rate intervals (bins). In the analysis we have used
rain rate bins of variable size to obtain a meaningful number of pixels within each bin.
These variables are defined as

$$Relative\ bias\ \% = 100 * \frac{\sum_{i=1}^{N}(mw_i - rr_i)}{\sum_{i=1}^{N} rr_i}$$





$$Adjusted\ FSE\% = \frac{\sqrt{\frac{1}{N}\sum_{i=1}^{N}(mw_i - rr_i - bias)^2}}{\frac{1}{N}\sum_i rr_i} * 100$$

where $mw_i$ is the PNPR (v1 or v2) rainfall rate and $rr_i$ is the TRMM-PR rainfall rate. N represents
the number of pixels in each precipitation rate bin.
Considering the $RB_\%$ (top panel of Fig. 8) it is evident the better performance of PNPR v2 in the
rain rate estimation over ocean (solid blue line) with respect to the PNPR v1 (dashed blue line). The
high relative bias over ocean for low rain rates in the PNPR v1 is significantly reduced in PNPR v2.
In the interval between 0 and 4 mm h$^{-1}$ the $RB_\%$ ranges from 300% to 100% for the PNPR v1 and
from 50% to 1% for the PNPR v2. In the following interval (4 to 10 mm h$^{-1}$) the values varies from
-5% to -40% for both the algorithms with a slightly better performance of PNPR v2.
Over land both the algorithms present similar performances, with a slightly better result for PNPR
v2 (solid black line) for low rain rates (0 - 3 mm h$^{-1}$), and lower $RB_\%$ of PNPR v1 (dashed black
line) for higher rain rate values (> 3 mm h$^{-1}$).
In the bottom panel of Fig. 8 $AFSE_\%$ shows high values (> 250%) over ocean (blue curves) for very
low rain rates (< 0.5 mm h$^{-1}$) in PNPR v1, while for PNPR v2 $AFSE_\%$ is much lower (< 200%). For
higher rain rates the two curves are similar, with slightly better performance of PNPR v2 for rain
rates < 2 mm h$^{-1}$, and better performance of PNPR v1 for rain rates between 2 mm h$^{-1}$ and 6 mm h$^{-1}$.
Over land, PNPR v2 shows lower $AFSE_\%$ compared to PNPR v1 for mid to high rainfall rates (> 2
mm h$^{-1}$), while for lower rain rates the two curves are similar, with slightly better results for PNPR
v2 for very low rain rates (< 0.5 mm h$^{-1}$), and for PNPR v1 for rain rates between 0.5 mm h$^{-1}$ and 2
mm h$^{-1}$. It should be noted that overall this comparison shows a general agreement in the capability
to estimate the precipitation by the two algorithms (as expected since they are based on the same
physical foundation) with better performance of the ATMS version of PNPR for low precipitation
rates, in particular over ocean (both in terms of $RB_\%$ and $AFSE_\%$). For higher precipitation rates
both PNPR versions tend to underestimate the precipitation (negative $RB_\%$), with larger (negative)
bias of PNPR v2 than PNPR v1 over land.
It is worth noting that the main improvement of PNPR v2 with respect to PNPR v1 is the reduction
of the relative bias ($RB_\%$) for low precipitation rates (where $RB_\%$ was higher) especially over ocean.
Moreover, considering the $AFSE_\%$, the error is prevalently lower in version 2 even if the effect of
the bias reduction is not taken into account.
**5. Summary and conclusion**
This paper describes the design of a new algorithm, PNPR v2, for estimation of precipitation on the
ground for the cross-track ATMS radiometer, and presents the results of a verification study where
the instantaneous precipitation rate estimates available from TRMM and GPM spaceborne radars
are used as reference.
PNPR v2 has been designed for retrieval of precipitation in the MSG full disk area. The algorithm,
based on a neural network approach, represents an evolution of the previous version PNPR v1,
designed for the AMSU/MHS radiometer, with some changes made to take advantage of the
improvements of ATMS with respect to AMSU/MHS. Similarly to the previous algorithm it is
based on a single neural network for all types of surface background, trained using a large database
based on 94 cloud-resolving model simulations over the European and the African areas.
The verification study carried out through a comparison with co-located observations of ATMS
with the NASA/JAXA TRMM-PR and GPM-KuPR spaceborne radars analyzed on a 0.5° × 0.5°
regular grid showed a substantial agreement of PNPR v2 with the precipitation products available
from the two radars.  In the comparison with TRMM-PR, over the African area, the CC has values
between 0.64 (arid land) and 0.71 (vegetated land), and RMSE varies between 0.80 mm h$^{-1}$ (arid





land) and 1.37 mm h$^{-1}$ (coast). The Adjusted FSE$_{\%}$, as a function of PR precipitation rate, ranges from 250% to 130% over ocean and from 250% to 100% over land in the interval from 0.1 to 1 mm h$^{-1}$. It is less than 50% for rain rate greater than 7 mm h$^{-1}$ (ocean and land). In the comparison with GPM-KuPR over the European area the indexes are quite comparable with those over the African area, with lower correlation (0.59 over vegetated land and 0.57 over ocean) and RMSE (0.82 mm h$^{-1}$ over vegetated land and 0.71 mm h$^{-1}$ over ocean). It is worth noting that the study based on GPM-KuPR will be further developed in the future using a larger coincidence dataset and a more consolidated version of DPR precipitation products.

For reference, it is useful to compare these results with those found by other authors carrying out validation studies using ground-based radar data. Tang et al. (2014) investigated the performance of PMW precipitation products from 12 passive microwave radiometers, including AMSU-B (NOAA 15, 16, 17) and MHS (NOAA 18, 19 and MetOp-A) rainfall rate estimates based on Ferraro et al. (2005) over a 3 year period. They found values of CC around 0.55 (on an annual scale, at 0.25° × 0.25° regular grid) over the continental United States (land). They also analyzed the adjusted root-mean-square errors normalized with the precipitation rate (corresponding to the Adjusted FSE% used in this study) as a function of ground radar precipitation rate, and have found values ranging from about 600% at 0.25 mm h$^{-1}$ to about 75% in the interval 6-16 mm h$^{-1}$ for winter, and from about 1200% at 0.25 mm h$^{-1}$ to about 50% above the 10 mm h$^{-1}$ for summer. Kidd et al. (2016) have analyzed the performance of precipitation retrieval from the MHS of the NASA Goddard PROFiling (GPROF) algorithm version developed for cross-track PMW sensors. Using quality controlled ground-based radar data over the United States from 6 March 2014 through 5 March 2015, and computing the statistical scores over a 1° x 1° grid, they have found CC < 0.50 over the western U.S., and > 0.60 over the eastern U.S. It is worth noting that a comparison of the measured performances of different algorithms is very difficult and may not be significant if the conditions in which the various studies are performed are different (e.g. for the type and the quality of the reference data, different climate regimes, different matching procedure, and spatial resolution used in the analysis). Therefore, what can emerge from such results is that PNPR v2 performance is at least comparable with those of the analyzed algorithms.

In the comparison of PNPR v2 and PNPR v1 retrievals, performed over the African area and based on two years period of coincident observations of ATMS and AMSU/MHS radiometers with TRMM-PR, an appreciably better performance of PNPR v2 has been evidenced by statistical indexes (e.g. CC equal to 0.71 for PNPR v2, vs. 0.68 for PNPR v1 over vegetated land, and equal to 0.69 for PNPR v2 vs. 0.61 for PNPR v1 over ocean), and by a general improvement of the estimate of low precipitation, mostly over ocean. The resulting differences can likely be attributed to improvements in the design of the neural network and also to the best technical features of ATMS compared to AMSU/MHS.

Overall, the two versions of PNPR algorithm have shown a general consistency in the results, as expected considering that both are based on the same physical basis (the training databases are based on the same cloud-resolving model and to the same radiative transfer model). It is worth noting that the achievement of consistency between products derived from different sensors is very relevant in the current GPM mission era, with constellation satellites (equipped with cross-track or conical scanning microwave radiometers) contributing to global coverage and higher temporal sampling of precipitation. This aspect has become very important also within the EUMETSAT H-SAF program, and represents a guideline for the development of PMW precipitation products. PNPR v2 and PNPR v1 for ATMS and AMSU/MHS, as well as other products for conically scanning radiometers (e.g. CDRD for SSMIS – Casella et al., 2013, Sanò et al., 2013), and new products for the other constellation radiometers are developed within H-SAF in this direction, with foreseen improvements of derived MW/IR products used in operational hydrology and near real time precipitation monitoring applications.

The results, however, have revealed a slight tendency of PNPR v2 to underestimate moderate to high precipitation, mostly over land, and overestimate moderate to light precipitation over the


ocean, especially compared to GPM-KuPR product over the North Atlantic Ocean. Besides well-known issues affecting PMW precipitation retrieval, such as non-uniform beam-filling effects related to small-scale rainfall structures associated with local convection and difficulties in the retrieval of warm or shallow rain processes, in addition to the lack of low frequency channels very useful for precipitation retrieval over ocean, other issues might be related to the use of spaceborne radar products as reference. The impact of sample volume discrepancies between radiometers and spaceborne radars, and uncertainties in the spaceborne radar estimates (due to attenuation correction, sensitivity thresholds, non-uniform beam filling effect), needs to be evaluated when using spaceborne radar precipitation estimates as reference. PNPR v2 will undergo thorough extensive validation within the EUMETSAT H-SAF program carried out by the H-SAF Precipitation Products Validation Service (Puca et al., 2014), using ground-based radars and rain gauges over Europe, and, in limited areas, over Africa, which will be useful to clarify some of these issues.

In spite of the above mentioned limitations, this study shows that the TRMM and GPM spaceborne radars can be very useful for an extensive verification, over long time periods, of consistency and accuracy of instantaneous precipitation rate estimates from different sensors. The use of spaceborne radars as reference overcomes some of the limitations in the use of ground-based data (such as inhomogeneity in their technical characteristics and data treatment, limited coverage, and beam blocking) providing consistent measurements around the globe, including remote areas where ground-based data are scarce or not available, and oceans.

**Acknowledgments**

The authors would like to thank the National Aeronautics and Space Administration (NASA, http://www.nasa.gov) for providing the TRMM-PR and GPM-KuPR data, the National Oceanic and Atmospheric Administration (NOAA, http://www.class.ngdc.noaa.gov) for providing the ATMS radiometer data, and The European Centre for Medium-Range Weather Forecasts (ECMWF, http://www.ecmwf.int) for providing the model reanalysis. This research was supported by EUMETSAT through the "Satellite Application Facility on Support to Operational Hydrology and Water Management" (H-SAF), by the Earth2Observe FP7 EU funded project, by the Italian Civil Protection Department. This research has been carried out within the collaboration between H-SAF and GPM (no-cost proposal approved by the NASA PMM Research Program) on the development of precipitation retrieval algorithms and validation activity.





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



Table 1 - Results of the tests for the selection of the inputs to the NN. Input combinations are listed
in the first column ($\Delta_F = \Delta_{13}, \Delta_{37}, \Delta_{17}$), R indicates the correlation coefficient, MSE the mean
squared error, CV the cross-validation index (subscripts L and CV indicate the learning and cross-
validation phases).

| INPUT | $R_L$ | $MSE_L$ | $R_{CV}$ | CV |
|---|---|---|---|---|
| **$\Delta_F$** | **0.85** | **0.39** | **0.76** | **0.42** |
| $\Delta_F$ and $\Delta_{14}, \Delta_{24}, \Delta_{27}$ | 0.78 | 0.64 | 0.70 | 0.68 |
| $\Delta_F$ and $\Delta_{14}, \Delta_{24}$ | 0.89 | 0.37 | 0.80 | 0.42 |
| $\Delta_F$ and $\Delta_{14}, \Delta_{27}$ | 0.83 | 0.49 | 0.78 | 0.53 |
| $\Delta_F$ and $\Delta_{24}, \Delta_{27}$ | 0.81 | 0.50 | 0.70 | 0.54 |
| $\Delta_F$ and $\Delta_{14}$ | 0.87 | 0.37 | 0.79 | 0.41 |
| **$\Delta_F$ and $\Delta_{24}$** | **0.92** | **0.32** | **0.87** | **0.35** |
| $\Delta_F$ and $\Delta_{27}$ | 0.83 | 0.48 | 0.68 | 0.52 |



Table 2 -  Contingency table of PNPR v2 retrievals relative to TRMM-PR measurements

| | Radar rain rate (mm h⁻¹) | | | |
|---|---|---|---|---|
| **Vegetated land** | **0.01 ≤ Rad ≤ 0.25** | **0.25 < Rad ≤ 1.0** | **1.0 < Rad ≤ 5.0** | **5.0 < Rad ≤ 15.0** |
| **0.01 ≤ PNPR v2 ≤ 0.25** | **80.3%** | 35.4% | 2.9% | 0.0% |
| **0.25 < PNPR v2 ≤ 1.0** | 17.8% | **50.4%** | 35.1% | 6.2% |
| **1.0 < PNPR v2 ≤ 10.0** | 1.9% | 13.6% | **54.6%** | 38.4% |
| **5.0 < PNPR v2 ≤ 15.0** | 0.0% | 0.6% | 7.4% | **55.4%** |
| **Coast** | **0.01 ≤ Rad ≤ 0.25** | **0.25 < Rad ≤ 1.0** | **1.0 < Rad ≤ 5.0** | **5.0 < Rad ≤ 15.0** |
| **0.01 ≤ PNPR v2 ≤ 0.25** | **86.4%** | 34,0% | 7.1% | 0.0% |
| **0.25 < PNPR v2 ≤ 1.0** | 12.1% | **52,3%** | 31.1% | 0.0% |
| **1.0 < PNPR v2 ≤ 10.0** | 1.5% | 13.0% | **51,0%** | 26.6% |
| **5.0 < PNPR v2 ≤ 15.0** | 0.0% | 0.7% | 10.8% | **73.4%** |
| **Ocean** | **0.01 ≤ Rad ≤ 0.25** | **0.25 < Rad ≤ 1.0** | **1.0 < Rad ≤ 5.0** | **5.0 < Rad ≤ 15.0** |
| **0.01 ≤ PNPR v2 ≤ 0.25** | **85.2%** | 32.5% | 3.7% | 0.0% |
| **0.25 < PNPR v2 ≤ 1.0** | 13.1% | **51.4%** | 30.8% | 7.5% |
| **1.0 < PNPR v2 ≤ 10.0** | 1.7 | 16.0% | **54.7%** | 39.1% |
| **5.0 < PNPR v2 ≤ 15.0** | 0.0% | 0.1% | 10.8% | **53.4%** |
| **Arid land** | **0.01 ≤ Rad ≤ 0.25** | **0.25 < Rad ≤ 1.0** | **1.0 < Rad ≤ 5.0** | **5.0 < Rad ≤ 15.0** |
| **0.01 ≤ PNPR v2 ≤ 0.25** | **72.1%** | 27.6% | 1.2% | n/a |
| **0.25 < PNPR v2 ≤ 1.0** | 27.6% | **53.4%** | 37.1% | n/a |
| **1.0 < PNPR v2 ≤ 10.0** | 0.3% | 18.7% | **58.0%** | n/a |
| **5.0 < PNPR v2 ≤ 15.0** | 0.0% | 0.3% | 3.7% | n/a |

(PNPR v2 rain rate (mm h⁻¹))




2 Table 3 -  Performace indexes for the different background surfaces

| | Vegetated land | Ocean | Coast | Arid land |
|---|---|---|---|---|
| Perf. Index | 69.0% | 76.0% | 75.7% | 66.7% |



1  Table 4 - Statistical indexes obtained in the comparisons of PNPR v2 retrievals with GPM-KuPR
2  and TRMM-PR products

|  | Vegetated land GPM-KuPR /TRMM-PR | Ocean GPM-KuPR /TRMM-PR |
|---|---|---|
| BIAS (mm h$^{-1}$) | 0.15/-0.08 | -0.04/0.05 |
| CC | 0.70/0.71 | 0.70/0.69 |
| RMSE (mm h$^{-1}$) | 1.08/1.11 | 1.21/1.30 |



1    Table 5– Statistical indexes of the comparison of PNPR v1 and PNPR v2 vs TRMM-PR retrievals

| | Arid land PNPR v1/PNPR v2 | Vegetated land PNPR v1/PNPR v2 | Coast PNPR v1/PNPR v2 | Ocean PNPR v1/PNPR v2 |
|---|---|---|---|---|
| BIAS (mm h$^{-1}$) | 0.30/-0.05 | -0.09/-0.08 | 0.15/0.05 | 0.63/0.05 |
| CC | 0.66/0.64 | 0.68/0.71 | 0.75/0.65 | 0.60/0.69 |
| RMSE (mm h$^{-1}$) | 1.84/0.80 | 1.66/1.11 | 1.73/1.37 | 2.25/1.30 |




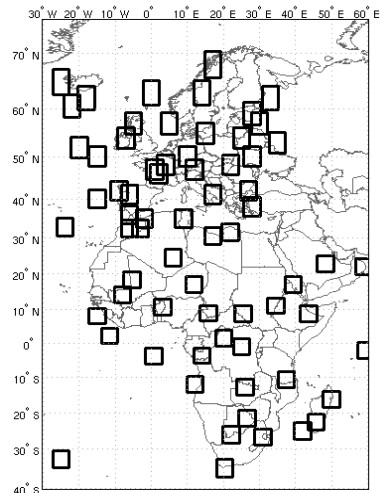

Figure 1 – Geographical location of the inner domain of the 94 NMS simulations over European
and African areas.

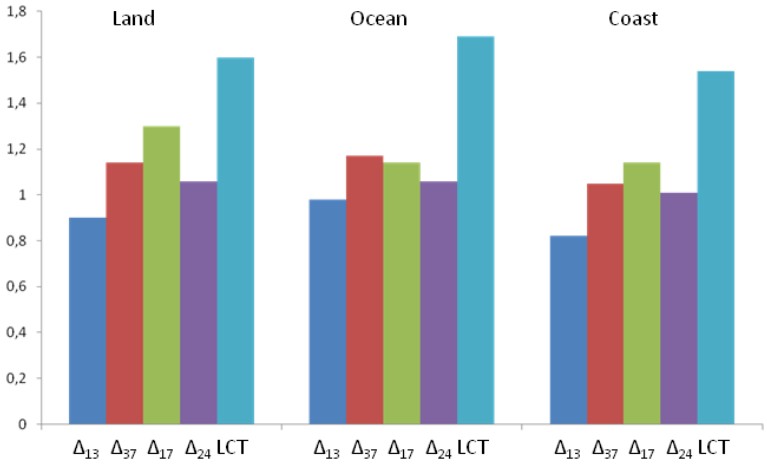

Figure 2  Relative sensitivity ($S_i$) of the NN evaluated for five inputs ($\Delta_{13}$, $\Delta_{37}$, $\Delta_{17}$, $\Delta_{24}$, LCT), for
three different background surfaces.




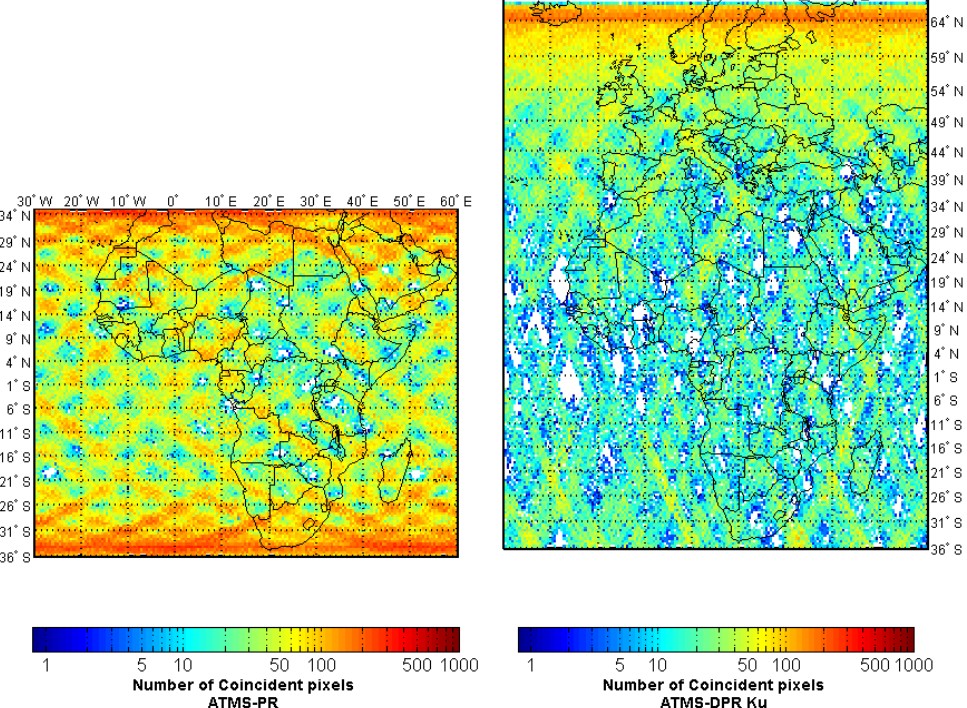

Figure 3 - Number of co-located pixels from TRMM-PR and the Suomi-NPP ATMS coincident
overpasses over the African area in the 24 month period 2013-2014 (left panel), and from GPM-Ku-
NS and Suomi-NPP ATMS coincident overpasses over European and African areas in the 15 month
period (March 2014 - May 2015) (right panel).





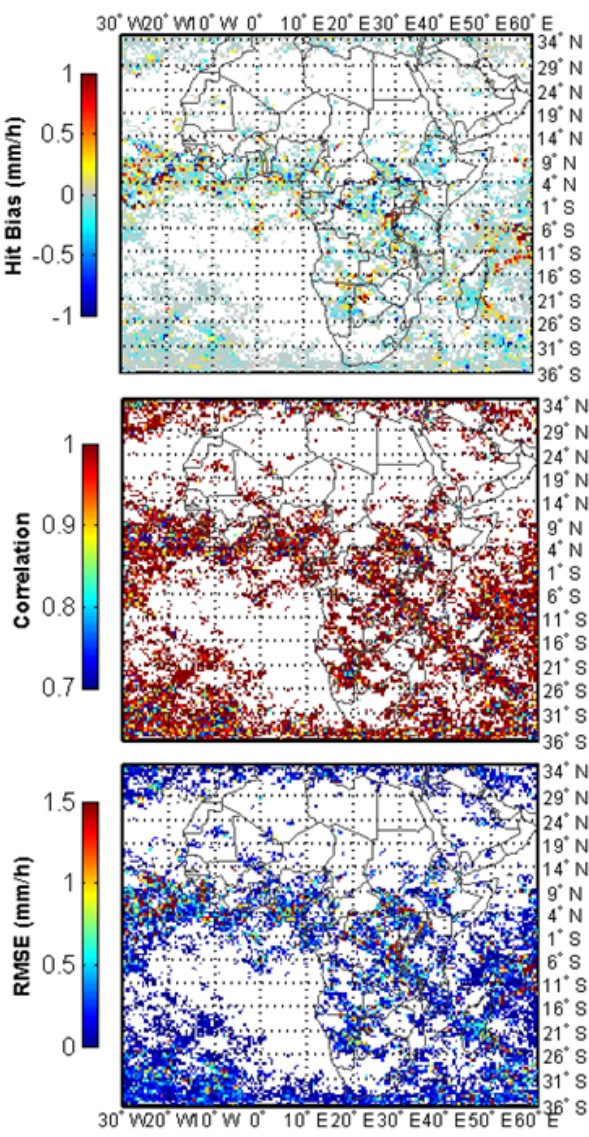

Figure 4 - Hit bias (top panel), correlation coefficient (CC, middle panel) and root mean squared
error (RMSE, bottom panel), resulting from the comparison between PNPR v2 and TRMM-PR
retrievals over the African area.



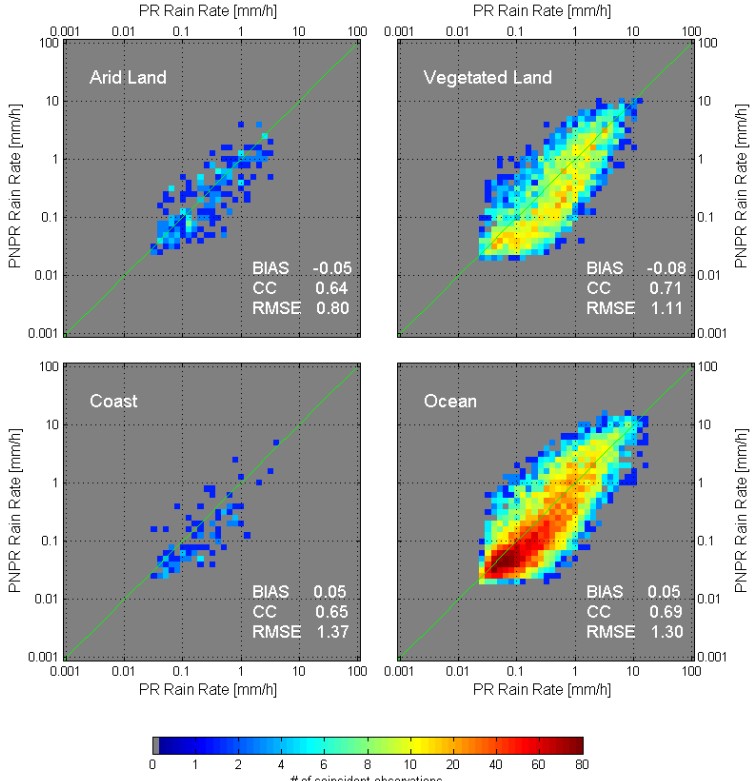

Figure 5 - Density scatterplots of the PNPR v2 and TRMM-PR mean rainfall rates (over a 0.5° × 0.5° regular grid), for the African area, for different surface types. A logarithmic scale is used for the precipitation rates in mm h$^{-1}$.





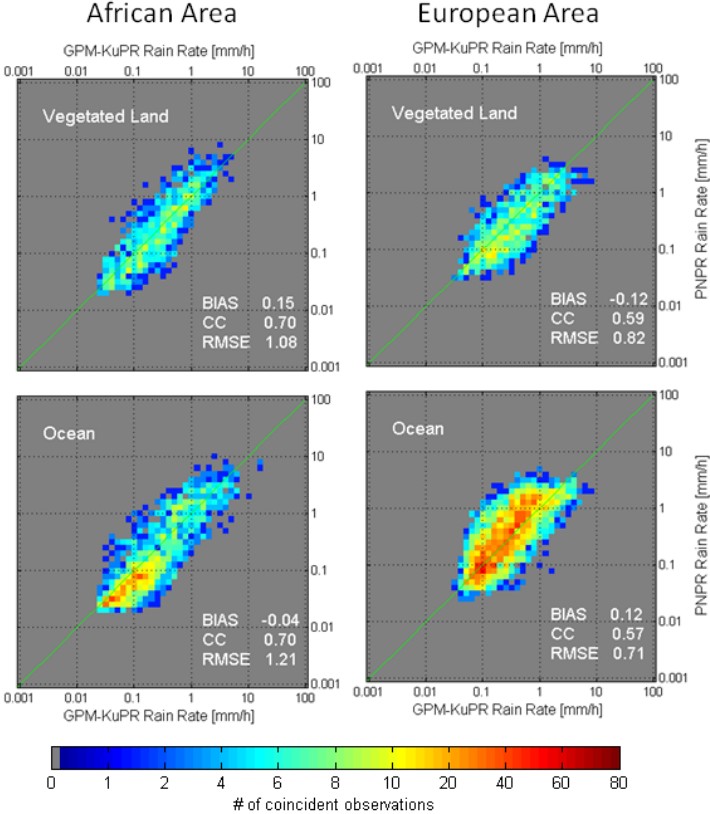

Figure 6 – Density scatterplots of the PNPR v2 and GPM-KuPR mean rainfall rates (over a 0.5° ×
0.5° regular grid), for the African area (left panels), and the European area (right panels), for
vegetated land and ocean. A logarithmic scale is used for the precipitation rates in mm h$^{-1}$.

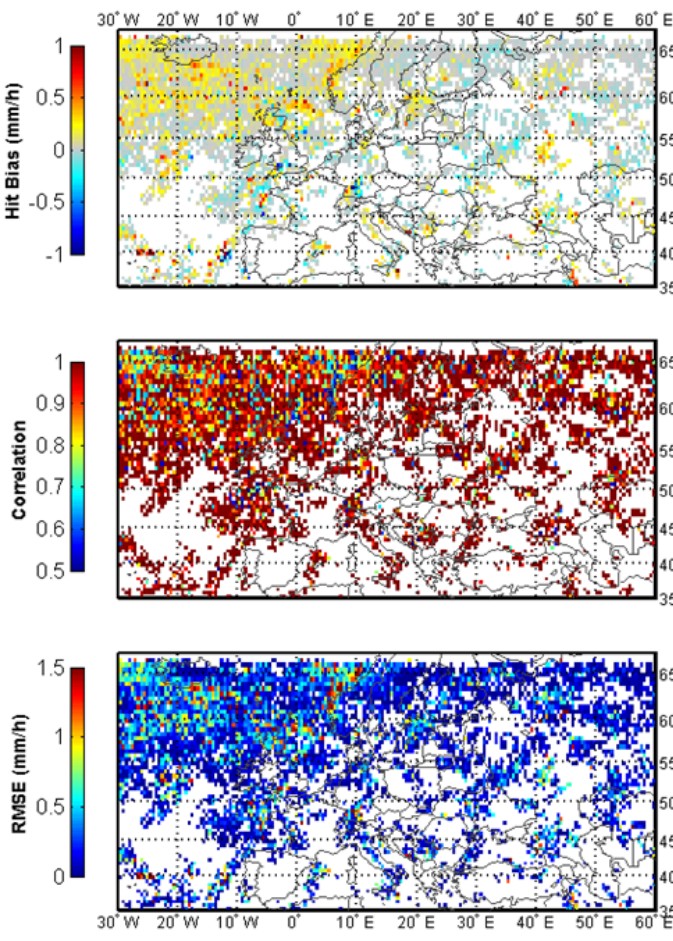

Figure 7 - Hit bias (top panel), CC (middle panel) and RMSE (bottom panel), resulting from the
comparison between PNPR v2 retrievals and GPM-KuPR measurements over the European area.



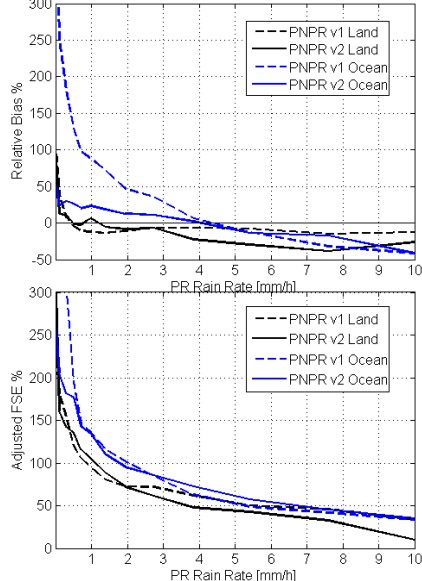

Figure 8 – Relative Bias percentage (top panel) and Adjusted FSE percentage (bottom panel), of
PNPR v1 and PNPR v2 retrievals with respect to the TRMM-PR measurements.
