# Peer review of "The new Passive microwave Neural network Precipitation Retrieval (PNPR)"

_Atmospheric Measurement Techniques, 2016_

## Referee Comment (RC1) · Anonymous Referee #1 · 19 Jul 2016

The paper presents an updated/new version of the H-SAF neural network precipitation retrieval scheme. The paper is informative, describing in detail the formulation and operation of the scheme, and presents some results in order to assess its performance. Overall the paper is well written, although there are a few typographical errors that should be addressed (see below).

P3, L41: 'Could' should be 'Cloud' P5, L35 and P5, L46: correct the 'spatial sampling of 1.11' to 'sampled every 1.11' – it is an angular measurement, not spatial. P7, section

3.2: it may be useful to clarify whether the training database was generated using system-specific simulations, or random. This is somewhat critical since is system-specific then the database could have a wet-weather bias. Also, was the PR data used in the database? I presume so, in which case this does explain some of the good performance against the PR later on. P8, L9: while the 183+-7 GHz can be sensitive to the surface, it should be noted that this channel (and neighbouring channels) are essential for near surface precipitation. P10, L23-25: the GPM DPR has the same swath width as the TRMM PR. P11, section 4.2: I didn't find at any stage a cautionary note that the satellite radars are insensitive to light precipitation - PR is essentially insensitive to rain intensities <0.7 mm/hr, the DPR c. <0.5 mm/hr. P15, L22-23: the correlations mentioned from Kidd et al 2016 are at 15 km resolution – the plotted data is summarized at 1 degree resolution.

Tables/Figures: Table 2: useful to include the spatial resolution in the caption. Figure 1: Are the orientations of the individual boxes the correct way around; I would expect that as latitude extends away from the Equator that the plotted x-dimension of the box would increase compared to the y-dimension. Figure 4/7: Include dates/resolution in the caption – and might be worth considering changing the resolution since the images are currently very noisy. Figure 5/6: Remove the 0.001-0.01 part of the plots (since there is no data in this region!).

---

## Referee Comment (RC2) · Anonymous Referee #2 · 29 Jul 2016

This study introduced a new version of the Passive microwave Neural network Precipitation Retrieval (PNPR v2), an algorithm for estimating instantaneous surface precipitation The main improvement of PNPR v2 in comparison with PNPR v1 contains several parts. Firstly, new data source from ATMS radiometer, which is more advanced in resolution and channel numbers than its predecessors AMSU and MHS, was used. Secondly, some new input variations were considered, especially the difference between the TBs of new channels ($183.31\pm4.5$GHz and $183.31\pm1.8$GHz). Thirdly, PNPR v2 designed one unique NN that is capable of operating on the whole area regardless of the type of surface and geographical area. The new PNPR v2 performs well benchmarked by TRMM-PR and GPM-KuPR. The paper also introduced potential contribution to the GPM mission which is attractive and inspiring. In general, the expression of this paper is good. The author reached all parts of a good paper required. Some revision is required before the paper is accepted.

Major Comments: 1. As the main body of PNPR v2 is very similar with PNPR v1, which is carefully described in another paper of the author in 2015, the author gave little technical description of the algorithm. But for completeness, the major technical parts of the algorithm should still be introduced with formula or figures briefly (e.g. the set of NN, the method to update weights and so on). It will facilitate readers to avoid additional literature search.

2. In page 6 and line 48, the author said the phase of the precipitation (solid, liquid, mixed or unknown) is contained in output. Is there any result and analysis of that? DPR can also differentiate solid and liquid phase.

3. In page 8 and line 6, the author used more than one hundred NNs to select the optimal network. What's the principle in adjusting the networks to get closer to the better one? And how can you tell the present one is the most optimized with some criteria?

Minor Comments: 1. In page 10 and line 37, "15 minute" should better be "15-minute".

2. In page 11 and line 36, there should be a comma at the end of "and over vegetated land (for all precipitation rates)". There are some other places where a comma is missing, the author should check by yourself again. 3. It would be better if Figure 1 is going to be turned into color one.

4. The numbers on the diagonal of Arid land in Table 2 were not bold.

5. In page 7 and line 20, how to define one "entry" and one "view"? What are two million

entries and 45 views consist of? The training database covers different seasons and different meteorological situations and precipitation regimes, is the number of each season, situation and regime equal to others?

6. In page 10 and line 37, what does "within a 15 minute time window" means?

7. In Page 11 and line 50, how to determine the intervals (0.01 - 0.25 mm/h, 0.25 - 1 mm/h, 1 - 5 mm/h and 5 - 15 mm/h)?

8. In Figure 3 (left one) and Figure 4, why the pixels presented in Figure 4 isn't consistent with the distribution of number of coincident pixels in Figure 3?
* * *

---

## Author Comment (AC1) · 7 Sep 2016

We would like to thank Referee#1 for his/her review of our paper and the important comments and suggestions provided. Please, find below our responses to the Referee's comments and the details on how we will address them in the new version of the manuscript.

**1. Referee comment: P3, L41: 'Could' should be 'Cloud'**

Authors' Reply: the sentences will be changed in the revised manuscript.

**2. Referee comment:** P5 L35 and P5 L46: correct the 'spatial sampling of 1.11' to 'sampled every 1.11' – it is an angular measurement, not spatial.

Authors' Reply: the sentences will be changed in the revised manuscript.

**3.1 Referee comment:** P7, section 3.2: it may be useful to clarify whether the training database was generated using system-specific simulations, or random. This is somewhat critical since is system specific then the database could have a wet-weather bias.

**Authors' Reply:**

The training database was generated using simulation of different precipitation events "in order to cover the different seasons and different meteorological situations and precipitation regimes". The selection of the simulations in terms of season, typology of event and geographical location was performed in order to optimize the completeness and representativeness of the database for the area of interest (see Casella et al., 2013). In detail, over the European/Mediterranean area we have considered 15 different meteorological events with different precipitation regimes for each season over different geographical areas. In the simulations over the African and South Atlantic area we have considered the different climatic regions and a sufficient number of simulations in order to obtain a reliable representation of the climate variability of each region. For a detailed list of the simulated events please see Answer (5) to Referee 2.

In the training phase we have considered only the elements of the database with a corresponding rain rate greater than 0 mm/h. The problem of the wet-weather bias has been faced using, in the retrieval process, a screening procedure in order to eliminate the no-rain pixels (see Sanò et al., 2015).

According to the Referee's suggestion, the sentence will be changed to clarify this point in section 3.2 of the revised version of the manuscript.

Revised version (Section 3.2 "The training database", lines 12-18, pag. 7):

Simulated events were selected in order to cover the different seasons and different meteorological situations and precipitation regimes. The selection of the simulations in terms of season, typology of event and geographical location was performed in order to optimize the completeness and representativeness of the database for the area of interest (see Casella et al., 2013). In detail, over the European/Mediterranean area we have considered 15 different meteorological events for each season over different geographical areas. Simulations over African and South Atlantic area were chosen also on the basis of the TRMM-PR observations (in particular the Rain Type flag and the Freezing level height) and on the basis of different climatic regions in order to cover as much as possible the climatic variability in the area of interest with a limited number of simulations.

**3.2 Referee comment:** Also, was the PR data used in the database? I presume so, in which case this does explain some of the good performance against the PR later on.

**Authors' Reply:**

As detailed in previous reply, the training database has been created using only simulated data. The TRMM-PR and GPM-Ku band radar data were not included in the database and they have been used as reference in the verification study.

**4. Referee comment:** P8, L9: while the 183+-7 GHz can be sensitive to the surface, it should be noted that this channel (and neighbouring channels) are essential for near surface precipitation.

**Authors' Reply:**

We agree with the Referee. The 183+-7 GHz and neighbouring channels are essential to estimate the near surface precipitation as mentioned in section 3.1, P6 L24-32, section 3.3, P7 L25-30, and in fact their contribution is an important part of the retrieval algorithm (inputs 2-5 of the NN, P8 L 47-50). The 183+-7 GHz channel, however, can be strongly affected by the signal due to surface emission in cold and dry conditions reducing the accuracy of surface rain rate estimation. The meaning of the sentence P8, L9 is that the new input  $\Delta_{24}$  can drive the NN to detect these conditions, and to obtain more reliable rain rate estimation.

5. Referee comment: P10, L23-25: the GPM DPR has the same swath width as the TRMM PR.

**Authors' Reply:**

In the verification study of PNPR v2 we chose to use the GPM-KuPR as its characteristics are similar to those of the TRMM-PR. In the sentence we meant that the GPM-KuPR, similar to the TRMM PR, has a wider swath than the GPM-KaPR band radar.

**We will rephrase the sentence as follows:**

In this study we have considered only the GPM-KuPR products because of the similarity with the TRMM-PR and because its larger swath size compared to the GPM-KaPR offers better chances to find coincident observations with ATMS"

**6. Referee comment:** P11, section 4.2: I didn't find at any stage a cautionary note that the satellite radars are insensitive to light precipitation - PR is essentially insensitive to rain intensities <0.7 mm/hr, the DPR c. <0.5 mm/hr.

**Authors' Reply:**

According to the Referee's suggestion, a sentence will be added to clarify this point in section 5 "Summary and conclusion" of the revised version of the manuscript.

**Revised version (Section 5 "Summary and conclusion", P15 line 8):**

" It should also be noted that the results presented in this study may be affected by the low sensitivity of spaceborne precipitation radars to light precipitation. This aspect will be further investigated through validation procedures based on ground radars and rain gauges (i.e., Puca et al., 2014), and further studies and dedicated activities are foreseen on these important aspects within the on-going scientific collaboration between the EUMETSAT H-SAF and the NASA/JAXA PMM Research Program."

**7. Referee comment:** P15, L22-23: the correlations mentioned from Kidd et al 2016 are at 15 km resolution – the plotted data is summarized at 1 degree resolution.

Authors' Reply:

We thank the referee for pointing this out. The sentence will be rephrased in the revised manuscript as follows:

Revised version (Section 5 "Summary and conclusion", lines 18-23, pag. 15):

Kidd et al. (2016) have analyzed the performance of precipitation retrieval of the NASA Goddard PROFiling (GPROF) algorithm version developed for cross-track PMW sensors (MHS). Using quality controlled ground-based radar data over the United States from 6 March 2014 through 5 March 2015, and computing the statistical scores (at the native - 15.88 km x 15.88 km - retrieval resolution) over a 1° x 1° grid, they have found CC < 0.50 over the western U.S., and > 0.60 over the eastern U.S.

**Tables/Figures:**

**8. Referee comment:** Table 2: useful to include the spatial resolution in the caption.

Authors' Reply:

The suggestion is accepted. The caption will be rephrased in the revised manuscript.

**9. Referee comment:** Figure1: Are the orientations of the individual boxes the correct way around; I would expect that as latitude extends away from the Equator that the plotted x-dimension of the box would increase compared to the y-dimension.

Authors' Reply: We agree with the Referee. We modified the figure and we also added some details concerning the simulated events. The figure in the revised manuscript will be the following:

Figure 1

**9. Referee comment:** Figure 4/7: Include dates/resolution in the caption – and might be worth considering changing the resolution since the images are currently very noisy.

**Authors' Reply:**

The suggestion is accepted. The captions will be rephrased in the revised manuscript.

The choice of the resolution used stems from a compromise between a detailed analysis of the results, more noisy, and an analysis using data averaged over a larger area that could hide some critical aspects concerning the performance of PNPR v2 (e.g. surface precipitation estimation over coastal area, or intense events with a limited spatial extension). The use of the same resolution at  $0.5^{\circ}x \ 0.5^{\circ}$  for statistics and figures is a good compromise to reduce problems related to geolocation and time synchronization between satellite and radar and leads to a more direct and consistent evaluation of the results. Based on these considerations, it seems more appropriate to maintain the current resolution despite it appears more noisy.

**10. Referee comment:** Figure 5/6: Remove the 0.001-0.01 part of the plots (since there is no data in this region!).

Authors' Reply: The suggestion is accepted. The plots will be updated in the revised manuscript

---

## Author Comment (AC2) · 7 Sep 2016

We would like to thank Referee#2 for his/her review of our paper and the important comments and suggestions provided. Please, find below our responses to the Referee's comments and the details on how we will address them in the new version of the manuscript.

**Major Comments:**

**1. Referee comment:** As the main body of PNPR v2 is very similar with PNPR v1, which is carefully described in another paper of the author in 2015, the author gave little technical description of the algorithm. But for completeness, the major technical parts of the algorithm should still be introduced with formula or figures briefly (e.g. the set of NN, the method to update weights and so on). It will facilitate readers to avoid additional literature search.

**Authors' Reply:**

The description in the paper of the PNPR v2 design methodology is concise as the design is similar to that already described in Sanò et al., 2015. But we agree that adding some details may facilitate the understanding of the paper. To clarify this point a short paragraph containing some technical parts of the algorithm will be added in the manuscript. The figure of the NN diagram is shown in Figure 2 of Sanò et al. (2015) and is reported below for convenience.

**" 3.3 The neural network**

A detailed description of the NN is provided in Sanò et al. (2015), but some basic aspects are presented for completeness.

The neural network scheme, shown in Figure 2 in Sanò et al.(2015), is characterized by ni inputs, one input layer, two hidden layers, and a number of nodes for each layer (e.g. n1 for the first layer). Each node has its own transfer function and receives, as input, a weighted sum of the outputs of the previous layer. The output of the transfer function corresponds to the output of each node. For example, the output of a node (k-th),  $y_k$ , of the first hidden layer takes the form:

$$y_k(\omega, x) = f_2 \left[ \sum_{j=1}^{n_1} \omega_{kj} * f_1 * \left( \sum_{t=1}^{n_i} \omega_{jt} * x_t + b1 \right) + b2 \right]$$
(1)

where  $x_t$  are the input signals (*ni* values),  $\omega_{jt}$  are the weights connecting the inputs to the nodes of the input layer and  $\omega_{kj}$  the weights connecting the nodes of the input layer to the nodes of the first hidden layer,  $f_1$  and  $f_2$  are the transfer functions of the input layer and the first hidden layer, and  $b_1$  and  $b_2$  are the bias of nodes of the two layers. During the training phase (backpropagation network and Levenberg-Marquardt algorithm) a training database is used that provides the network with synthetic input and output data. The input signal propagates forward from the input layer of nodes to the output layer. The node in the output layer produces an output ( $y_i$ ), which is compared to the i-th target output ( $t_i$ ) defined in the training set. An error value is calculated as

$$E = \frac{1}{n} \sum_{i=1}^{n} (y_i - t_i)^2$$
(2)

where n is the number of elements of the training set. The network corrects its weights to lessen the errors. The iteration continues in order to minimize the error. At the end of the training phase the performance of the NN is measured by the mean squared error and the correlation coefficient.".

Schematic diagram of a multilayer neural network (two hidden layers) (from Sanò et al., 2015).

**2. Referee comment:** In page 6 and line 48, the author said the phase of the precipitation (solid, liquid, mixed or unknown) is contained in output. Is there any result and analysis of that? DPR can also differentiate solid and liquid phase.

**Authors' Reply:**

In the manuscript we have not provided any details about the procedure used in PNPR v2 to evaluate the phase of the precipitation as the procedure is the same used in PNPR v1, described in Sanò et al., 2015. In this study, we were focused mainly on the evaluation of the performance of the algorithm considering only liquid precipitation. We are currently carrying out a separate study on the discrimination the liquid/solid/mixed precipitation using ATMS (and GMI) measurements (and using Cloudsat and DPR as reference) in order to test and improve this procedure. The results will be presented in an upcoming paper.

**3.1 Referee comment:** In page 8 and line 6, the author used more than one hundred NNs to select the optimal network. What's the principle in adjusting the networks to get closer to the better one?

**Authors' Reply:**

The principle in adjusting the network includes two relatively distinct aspects: determining how many layers to use and determining how many nodes to include in each layer. A detailed description of the procedure is presented in Sanò et al. 2015, section 3.2, pag 841-842:

"the model selection has been carried out using a cross validation method (Anders and Korn, 1999; Marzban, 2009). In the cross validation strategy the comparison between two models is based on the mean square prediction errors (MSPE) which is obtained applying the model to different validation sets. For this purpose a test dataset is used, divided into M subsets containing n observations each. The model is repeatedly re-estimated using different dataset of n(M-1) observations, leaving out a different subset each time. The average MSPE defines the cross validation error, CV (Anders and Korn, 1999):

$$CV = \frac{1}{M} \sum_{m=1}^{M} MSPE_m$$

In the cross validation methodology, the first step consists in determining the number of hidden layers. Starting from a simple architecture, two models are compared, one of which contains an additional hidden unit. For both the models the CV is evaluated and, if the more complex unit shows a smaller CV error, the additional hidden layer is accepted. The procedure stops when no further hidden layer is able to reduce the CV error. At this point, with a similar procedure, the number of nodes is optimized in each layer. The second step aims at determining the input connections. To find irrelevant connection, one input is removed and the resultant CV is compared with that of the complete network. In this way all the models with one input connection removed are analyzed and the model with the lowest CV error is accepted. At the end of this second step, no input connection can be removed without increasing the CV error."

Due to the complexity of the procedure we considered appropriate to include in the manuscript just a reference to that paper.

**3.2 Referee comment:** In page 8 and line 6. And how can you tell the present one is the most optimized with some criteria?

Authors' Reply:

The criterion for determining the most optimized NN, as reported in pag. 8, line 40-43 is:

" "optimal" refers to the one with best performance, i.e., minimum CV over the full dynamic range of the inputs, absence of overfitting, and absence of anomalous inhomogeneities in the retrievals (Sanò et al., 2015; Staelin and Surussavadee, 2007)."

To clarify this point, this sentence will be moved to pag. 8 lines 4-7:

" It is worth noting that to achieve the results shown in Table 1 the training protocol described in Sanò et al. (2015) has been applied, and that for each input configuration (each row in the table) more than one hundred NNs (with different levels of perceptrons) were compared to select the optimal network configuration, where "optimal" refers to the one with best performance, i.e., minimum CV over the full dynamic range of the inputs, absence of overfitting, and absence of anomalous inhomogeneities in the retrievals (Staelin and Surussavadee, 2007)."

**Minor Comments:**

1. Referee comment: In page 10 and line 37, "15 minute" should better be "15-minute".

Authors' Reply: The sentences will be changed in the revised manuscript.

**2. Referee comment:** In page 11 and line 36, there should be a comma at the end of "and over vegetated land (for all precipitation rates)". There are some other places where a comma is missing, the author should check by yourself again.

Authors' Reply: We will review the manuscript to correct these errors.

**3. Referee comment:** It would be better if Figure 1 is going to be turned into color one.

Authors' Reply: The suggestion is accepted. The figures will be replaced in the revised version of the manuscript with the following: